# Susceptibility of *Amblyomma sculptum*, Vector of *Rickettsia rickettsii*, Ticks from a National Park and an Experimental Farm to Different Synthetic Acaricides

**DOI:** 10.3390/pathogens12111304

**Published:** 2023-10-31

**Authors:** Ennya Rafaella Neves Cardoso, Stephani Félix Carvalho, Sarah Alves Dias, Rayane Almeida Santos, Mariana Avelar Tavares, Lucianne Cardoso Neves, Warley Vieira de Freitas Paula, Gracielle Teles Pádua, Nicolas Jalowitzki de Lima, Raquel Loren dos Reis Paludo, Isabela Santos Silva, Raphaela Bueno Mendes Bittencourt, Gabriel Cândido dos Santos, Flavia Giovana de Jesus Nascimento, Luiza Gabriella Ferreira de Paula, Filipe Dantas-Torres, Caio Marcio De Oliveira Monteiro, Felipe da Silva Krawczak

**Affiliations:** 1Laboratório de Doenças Parasitárias—LADOPAR, Setor de Medicina Veterinária Preventiva, Escola de Veterinária e Zootecnia, Universidade Federal de Goiás—UFG, Goiânia 74690-900, Brazil; ennyaneves@discente.ufg.br (E.R.N.C.); stephani.carvalho@discente.ufg.br (S.F.C.); sarah_alves@discente.ufg.br (S.A.D.); rayalmeida@discente.ufg.br (R.A.S.); mariana.tavares@discente.ufg.br (M.A.T.); luciannecardoso@discente.ufg.br (L.C.N.); warleyvieira@discente.ufg.br (W.V.d.F.P.); gracielletelespadua@discente.ufg.br (G.T.P.); jalowitzki@discente.ufg.br (N.J.d.L.); raquelloren@unifimes.edu.br (R.L.d.R.P.); rafabmbitt@discente.ufg.br (R.B.M.B.); doscandido@discente.ufg.br (G.C.d.S.); flaviagiovana@discente.ufg.br (F.G.d.J.N.); luizadepaula@ufg.br (L.G.F.d.P.); 2Laboratório de Biologia, Ecologia e Controle de Carrapatos—LABEC, Centro de Parasitologia, Instituto de Patologia Tropical e Saúde Pública, Universidade Federal de Goiás—UFG, Goiânia 74690-900, Brazil; isabelasantoscbio@gmail.com (I.S.S.); caiosat@ufg.br (C.M.D.O.M.); 3Laboratory of Immunoparasitology, Department of Immunology, Aggeu Magalhães Institute, Oswaldo Cruz Foundation (FIOCRUZ), Recife 50740-465, Brazil; fdtvet@gmail.com

**Keywords:** *Amblyomma cajennense* complex, pyrethroids, amitraz, organophosphates, amidines, resistance

## Abstract

*Amblyomma sculptum* is a relevant tick species from a One Health perspective, playing an important role as a vector of *Rickettsia rickettsii*, the main agent of spotted fever rickettsiosis in Brazil. In this study, we evaluated the susceptibility of two *A. sculptum* populations from Goiás state (midwestern Brazil) to different acaricides. The first tick population (GYN strain) originated from an experimental farm, where the ticks are annually exposed to acaricides. The second (PNE strain) was collected in a national park (Emas National Park), where the ticks had not been exposed to acaricides. Immersion tests were conducted with 21-day-old laboratory-reared larvae and nymphs originating from adult ticks collected in the areas mentioned above. The chosen acaricides were two synthetic pyrethroids (cypermethrin and deltamethrin), one organophosphate (chlorfenvinphos), one formamidine (amitraz), and two combinations of pyrethroids and organophosphates (cypermethrin, chlorpyrifos and citronellal; cypermethrin, fenthion and chlorpyrifos). Mortality data were used to determine the lethal concentration (LC) values at which 50%, 90%, and 99% of the ticks died (LC_50_, LC_90_, and LC_99_, respectively), and resistance ratios (RR) were calculated based on the LC values. The RR revealed differences between the acaricide-exposed (GYN) and unexposed (PNE) tick strains. The PNE strain larvae and nymphs were susceptible to all the tested acaricides. The GYN strain larvae were tolerant to cypermethrin, whereas the nymphs were tolerant to deltamethrin, chlorfenvinphos, and the combination of cypermethrin, chlorpyrifos, and citronellal (2 < RR ≤ 10). The GYN strain nymphs were resistant to amitraz (RR > 10). This is the first report of *A. sculptum* nymphs with resistance to amitraz and tolerance to deltamethrin, chlorfenvinphos, and the combination of cypermethrin, chlorpyrifos, and citronellal.

## 1. Introduction

*Amblyomma sculptum* is a three-host ixodid tick, whose preferred hosts are horses (*Equus caballus*), capybaras (*Hydrochoerus hydrochaeris*), and tapirs (*Tapirus terrestris*) [1,2,3,4]. However, *A. sculptum* ticks are generalists, parasitizing wild and domestic animals, as well as humans. *Amblyomma sculptum* is present in different South American countries, such as Argentina, Bolivia, Brazil, and Paraguay [5,6]. These ticks are commonly found in tropical and subtropical moist broadleaf forests and grasslands savannas [6]. *Amblyomma sculptum*, which is a member of the *Amblyomma cajennense* complex, is the principal vector of *Rickettsia rickettsii*, the main agent of spotted fever rickettsiosis in Brazil [7,8], and can possibly transmit other pathogens to horses and other animals [9]. Therefore, its control is extremely important from a One Health perspective.

Studies have shown that larvae and nymphs of *A. sculptum* are more susceptible to acaracides than adults are [10,11]. Thus, the most common strategy to control *A. sculptum* in Brazil is the application of synthetic acaricides among horses between April and October [10,11], when the immature stages predominate [6]. The most frequently used acaricides on horses are pyrethroids, organophosphates and their combinations, but the emergence of acaricide resistance in horse ticks is a current concern [12].

The indiscriminate use of acaricides is a long-standing problem in tick control worldwide, with several reports of acaricide resistance in important tick species, such as *Rhipicephalus microplus* (cattle ticks) and *Rhipicephalus sanguineus* sensu lato (brown dog ticks) [13,14,15]. Unlike what happens with these ticks, the studies regarding the chemical control and acaricide resistance in horse ticks, such as *A. sculptum*, are limited. A recent study reported resistance to amidines and cypermethrin (synthetic pyrethroid) in the different life stages of *Amblyomma mixtum*, which also belongs to the *A. cajennense* complex [13]. In 2011, a study conducted in Goiás state, midwestern Brazil, reported resistance to deltamethrin (synthetic pyrethroid) in *A. sculptum* adults [16]. In this scenario, the purpose of this study was to evaluate the susceptibility of immature stages of two *A. sculptum* populations from Goiás. We were interested in comparing the susceptibility of two strains, one from a national park (with no previous exposure to acaricides) and another from an experimental farm (with previous exposure to acaricides).

## 2. Materials and Methods

### 2.1. Ethics Committee

The Ethics Committee on Animal Use of the Federal University of Goiás (CEUA/UFG) approved this study (protocol number 071/22), which was conducted in accordance with the ethical principles of animal experimentation. The Chico Mendes Institute for Biodiversity (ICMBio Permit No. 70143-2) authorized the collection of ticks in Emas National Park.

### 2.2. Study Locations

The study was carried out in Goiás state, whose climate is tropical, with two well-defined seasons; i.e., dry (May–September) and rainy seasons (October–April). Most of the state’s territory is situated within the Cerrado biome, the most diverse tropical savanna on Earth. The ticks were collected between October 2021 and October 2022 in two areas (Figure 1) where *A. sculptum* ticks are known to be present: Emas National Park (PNE) and the experimental farm of the School of Veterinary and Animal Science of the Federal University of Goiás (EVZ/UFG).

The PNE (17°53′47″ S, 53°0′17″ W) is a national park and an environmentally preserved area of the Cerrado biome. The park covers an area of approximately 132,000 ha, distributed across the municipalities of Mineiros, Chapadão do Céu, and part of Costa Rica (Mato Grosso do Sul), but the ticks were collected in Mineiros. Domestic animals are absent, and there is no history of acaricide treatment in this area. For this study, the ticks collected in the PNE (namely PNE strain) were considered to be the susceptible strain in the immersion tests for a statistical comparison with the ticks collected in the experimental farm (namely, the GYN strain). This experimental farm (16°35′42″ S, 49°16′50″ W) is in the municipality of Goiânia (capital of Goiás) (421.36 km from PNE) and consists of an area where chemical acaricides are often used on dairy cattle [17,18,19]. In addition to *R. microplus*, *A. sculptum* is commonly found on dairy cattle due to their close contact with capybaras. Dairy cattle and capybaras share a pasture during the night, and a high level of environmental infestation by *Amblyomma* ticks has been previously detected in this area using the dragging method [20].

### 2.3. Tick Colonies

Tick collections were conducted using the dragging method [20]. Dragging was conducted by passing a white flannel over ground-level vegetation. Adult ticks were collected with anatomical tweezers and stored in a 50 mL tube identified with date and location. At the Laboratory of Parasitic Diseases (LADOPAR) at EVZ/UFG, the ticks were kept in the BOD (Biochemical Oxygen Demand) incubator at 27 ± 1 °C and 80 ± 5% relative humidity (RH) until taxonomic identification was conducted using keys [21,22]. In total, 234 and 907 *A. sculptum* adults were collected in PNE and the experimental farm, respectively.

The adult ticks fed on rabbits (*Oryctolagus cuniculus*), one for each tick strain used in this study. The rabbits were kept in suitable cages under controlled environmental conditions (temperature of 20 °C) with water and food ad libitum. All tick infestations were conducted inside cotton sleeves glued to the rabbits’ shaved dorsum, as previously described [23,24]. Engorged females were collected daily and kept in a BOD incubator at 27 ± 1 °C and 80 ± 5% RH. Fifteen days after oviposition, the eggs were weighed and separated into portions of 0.3 g, which were added into 10 mL syringe barrels sealed with hydrophilic cotton plugs. These syringes were kept in the BOD incubator and inspected daily for egg hatching. To obtain nymphs, the larvae from each strain fed on guinea pigs (*Cavia porcellus*), following the same procedures as for rabbits.

### 2.4. Acaricides and Lethal Concentrations

Synthetic pyrethroids (deltamethrin and cypermethrin) and organophosphates (chlorfenvinphos) were chosen because they are commonly used for controlling *A. sculptum* on horses in Brazil. Combinations of pyrethroids and organophosphates (chlorpyrifos + cypermethrin + citronellal and chlorpyrifos + fenthion + cypermethrin) and amitraz were also included. Although they are not specifically indicated for horses, their inclusion was justified by the direct coexistence between capybaras and cattle in the experimental farm, where exposure to these acaricides occurs [17,18,19]. Indeed, acaricides, such as chlorpyrifos + high-cis cypermethrin, flumethrin, and chlorpyrifos + ethion + alpha-cypermethrin, have long been used to control *R. microplus* on dairy cattle in the experimental farm [17,18,19,25,26]. The selection of these acaricides was also relevant for comparison with a previous study conducted in Goiás [16].

Commercial products (Table 1) were directly purchased from consumer stores and diluted in distilled water following the manufacturer’s instructions. Briefly, after diluting the product as recommended by the manufacturer (i.e., first dilution), we prepared two-fold serial dilutions to obtain the concentration ranges described for the larvae in Table 1. For the nymphs, the concentration range was chosen based on results previously obtained with the larvae. Specifically, we chose concentrations that killed between 10% and 100% of the larvae.

The lethal concentration (LC) values for commercial products containing a single active ingredient were calculated based on the concentrations in parts per million (ppm). However, for acaricide combinations, the LC values were calculated using percentages of commercial acaricide. An initial concentration of 100% as indicated on the package leaflet was considered the reference point for these calculations [25].

### 2.5. Larval Immersion Tests (LITs)

LITs were performed according to the methodology described by Sabatini et al. [27]. Two independent assays were performed for each tick strain. The control group was treated with distilled water, with a maximum acceptable mortality rate of 5%. Approximately 100 unfed larvae (21 days old) were immersed in each acaricide solution in a microtube (2 mL) for 5 min. For each concentration, 10 replicates were performed, totaling 1000 larvae per concentration. After filtering the solution, the larvae were transferred to a clean, dry filter paper packet (6 cm × 6 cm) closed with metal clips. The packets were placed in a BOD incubator at 27 ± 1 °C and RH ≥ 80% for 24 h. After incubation, the dead and live larvae were counted. The tick larvae that did not move after the CO_2_ stimulus (by blowing on them) were considered dead. Mortality was calculated according to the formula: mortality (%) = dead larvae × 100/total larvae.

### 2.6. Nymphal Immersion Tests (NITs)

NITs were conducted as LITs were [27], with some modifications. The method of storing the engorged larvae for molting into nymphs was modified to facilitate the test execution. The engorged larvae were separated into groups of 120 individuals in 80 mL transparent plastic containers with the lids previously punctured with 25 mm × 0.7 mm needles. A diluted acaricide solution was added to the collection tube, and the nymphs (21 days old) were immersed in the solution with the aid of a wooden stick. After manual agitation for 5 min, the solution was filtered, and 10 nymphs were separated into 6 × 6 cm filter paper packets (with 10 replicates per concentration), which were sealed with metal clips. The control group was treated with distilled water, with a maximum acceptable mortality rate of 5% such as for the larvae. A mortality assessment after 24 h was performed using the same criteria as for LIT.

### 2.7. Statistical Analysis

The LC values at which 50%, 90% and 99% of the ticks died (LC_50_, LC_90_, and LC_99_, respectively) were determined using Probit analysis based on the mortality data obtained from LITs and NITs. Ninety-five confidence intervals (95% CI), chi-square (X^2^) and slope values were also calculated using RStudio software, version 4.3.1. For comparison between the GYN and PNE strains, resistance ratios (RR) were calculated using the LC values of the susceptible strain (i.e., RR_50_ = LC_50_ GYN/LC_50_ PNE; RR_90_ = LC_90_ GYN/LC_90_ PNE and RR_99_ = LC_99_ GYN/LC_99_ PNE). The ticks were categorized as susceptible (RR ≤ 2), tolerant (2 < RR ≤ 10), or resistant (RR > 10) based on the classification system established by Kaplan et al. [14]. The susceptibility assessment was determined by most of the calculated RR (RR_50_, RR_90_, and RR_99_). Additionally, overlapping confidence intervals between the strains were considered as statistically non-significant [14].

## 3. Results

### 3.1. LITs

The results of LITs with deltamethrin, cypermethrin, chlorfenvinphos, and amitraz with both *A. sculptum* strains are shown in Table 2 and Figure 2. The GYN strain larvae were susceptible to deltamethrin, chlorfenvinphos, and amitraz, but tolerant to cypermethrin. The highest mortality rate recorded in the larvae from the control group was 2%.

The GYN strain larvae were susceptible to the acaricide combinations tested (Table 3). All the LCs of chlorpyrifos + cypermethrin + fenthion were significantly higher for the GYN strain, as compared to those of the PNE strains. Yet, the LC_50_ obtained for chlorpyrifos + cypermethrin + citronellal was higher for the PNE strain than it was for the GYN strain (Figure 2E).

### 3.2. NITs

The results of the NITs with deltamethrin, cypermethrin, chlorfenvinphos, and amitraz with both *A. sculptum* strains are shown in Table 4 and Figure 3. The GYN nymphs were susceptible to cypermethrin, tolerant to deltamethrin and chlorfenvinphos, and resistant to amitraz. The LC_50_, LC_90_, and LC_99_ of amitraz were 56.2, 20, and 8.8 times higher, respectively, for GYN than those for PNE (Figure 3D). The highest mortality recorded in nymphs from the control group was 2.7%.

As shown in Table 5, the NITs indicated that the GYN strain nymphs were susceptible to the chlorpyrifos + cypermethrin + fenthion combination and tolerant to chlorpyrifos + cypermethrin + citronellal.

## 4. Discussion

We found that *A. sculptum* larvae originating from the adults collected in the experiment were tolerant to cypermethrin. Remarkably, the nymphs were resistant to amitraz and tolerant to deltamethrin, chlorfenvinphos, and to a combination of chlorpyrifos + cypermethrin + citronellal. This is the first report of *A. sculptum* resistance to amitraz and tolerance to chlorfenvinphos and to a combination of chlorpyrifos + cypermethrin + citronellal.

Two out of three previous studies investigating acaricide resistance in *A. sculptum* found high levels of susceptibility [10,28]. However, in a study in five different municipalities in the state (i.e., Caldas Novas, Hidrolândia, Goiás, Terezópolis, and Goiânia), Freitas et al. [16] reported resistance to deltamethrin in *A. sculptum* adults. According to the same authors, their results indicated a status of probable resistance to deltamethrin, cypermethrin + piperonyl butoxide, amitraz, and permethrin in the larvae. Our results did not confirm deltamethrin resistance in the studied *A. sculptum* populations, but indicate a tolerance to cypermethrin in the GYN strain. The finding of deltamethrin resistance by Freitas et al. [16] may be attributed to the fact that ticks used were collected from horses and had a greater exposure to this synthetic pyrethroid, which is the most common acaricide used on horses [9]. On the contrary, our ticks were collected from the environment in an experimental farm where ticks are less exposed to deltamethrin or other synthetic pyrethroids. It is also worth mentioning that Freitas et al. [16] did not report a resistance to amitraz, which we found in our study.

As expected, the LCs obtained for the PNE strain (no previous exposure to acaricides) were lower than those for the GYN strain (previously exposed to acaricides), with a few exceptions (see Table 3). The PNE strain is a preservation area without the circulation of domestic animals and without a history of acaricide treatment. Thus, it will provide a foundation for future studies and could be used as a susceptible reference strain.

The nymphs showed a tolerance to the combination of chlorpyrifos + cypermethrin + citronellal (RR > 2), which could possibly be explained by their tolerance to the individual classes of pyrethroids and organophosphates. Although these combinations are not licensed for use on horses according to the package insert, they have been used on dairy cattle in the experimental farm where the *A. sculptum* GYN strain was collected [17,18,19,25,26]. This may suggest that the constant use of acaricides targeting *R. microplus* on cattle may have resulted in the selection of a tolerant *A. sculptum* population. This is certainly a hypothesis that deserves further investigation.

In this regard, the higher frequency of acaricide resistance in cattle ticks as compared to that of horse ticks could be related to different factors, including the much higher frequency of acaricide application on cattle than that on horses [26]. Moreover, *R. microplus* is a one-host tick, and therefore, it remains in contact with the treated animals for longer as compared to *A. sculptum*, which is a three-host tick [6]. Finally, *R. microplus* completes up to five or six generations per year [29], which may favor the transfer of resistance genes to new generations.

Overall, higher acaricide concentrations were necessary to kill the nymphs as compared to the larvae in the present study. For example, a six times lower concentration of deltamethrin was sufficient to achieve 99% mortality in the larvae as compared to that of the nymphs. This natural resistance mechanism across different tick stages has been reported in a previous study, in which fed and unfed ticks were also compared [10]. The study showed that the fed ticks required even higher concentrations than the unfed ones, which could be attributed to the formation and thickness of the cuticles in different tick stages [10].

Amitraz is not recommended for horses due to its toxicity, which can lead to colic and even death [30]. The experimental farm where we collected adult ticks for testing is surrounded by rural properties, where capybaras roam freely. Local producers frequently use amitraz to control cattle ticks in the region. Our study is the first to document amitraz resistance in *A. sculptum* nymphs. The LC_50_ and LC_90_ of amitraz for the GYN strain (resistant) were 56.2 and 20 times greater, respectively, than those of the PNE strain (susceptible). In the only previous study assessing the susceptibility of *A. sculptum* to amitraz [16], the authors concluded that the larvae were probably resistant to this acaricide.

Ticks, such as *A. sculptum*, which are predominantly maintained by wildlife, but can also infest domestic animals and humans, are of great significance from a One Health perspective. Wild animals such as capybaras are coming into closer contact with humans due to the impact of anthropic actions on natural environments, and scenarios like the one described in this study, where capybaras and cattle share pastures, are becoming increasingly common [20]. This coexistence increases the risk of exposure to ticks and their associated pathogens for humans and other animals [31]. In a similar manner, this increases the exposure of ticks to acaricides that are not necessarily applied on their primary hosts (e.g., capybaras), but on domestic animals sharing the same environment. Ultimately, this is something to be considered while designing control strategies for certain tick species (e.g., *R. microplus*) in areas where other relevant tick species (e.g., *A. sculptum*) are present.

Considering the medico-veterinary significance of *A. sculptum*, further research is advocated to investigate whether other *A. sculptum* populations in Brazil are resistant to these acaricides and to ascertain the possible mechanisms underlying acaricide resistance in this tick species. In this perspective, it would be important to develop molecular methods for large-scale acaricide resistance surveillance in this tick species. This could include, for instance, PCR assays to detect single-nucleotide polymorphisms (SNPs) at target sites, such as the voltage-gated sodium channel for pyrethroids.

## 5. Conclusions

We conclude that *A. sculptum* nymphs from the experimental farm (i.e., previously exposed to acaricides) were resistant to amitraz and tolerant to deltamethrin, chlorfenvinphos, and to a combination of chlorpyrifos + cypermethrin + citronellal. The larvae were tolerant to cypermethrin. On the other hand, the larvae and nymphs originating from a preservation area (no previous contact with acaricides) were susceptible to all the acaricides tested.

## Figures and Tables

**Figure 1 pathogens-12-01304-f001:**
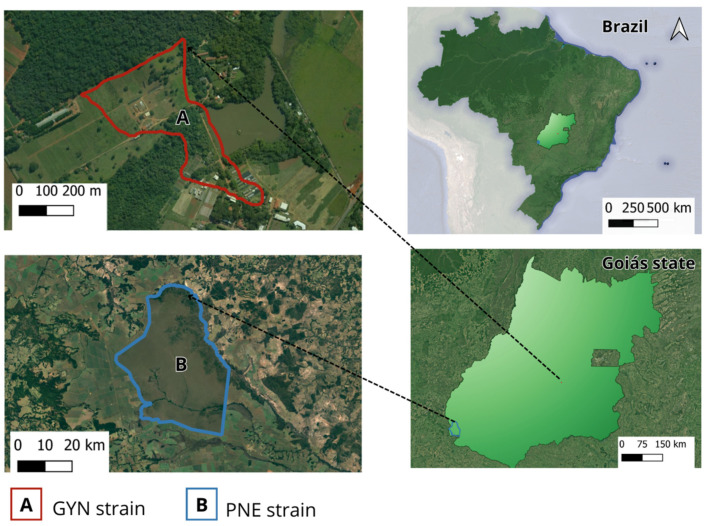
Location of tick collection sites in Goiás state (Cerrado biome), midwestern Brazil. (**A**) Experimental farm of the School of Veterinary and Animal Science of the Federal University of Goiás (EVZ/UFG) (red). (**B**) Emas National Park (blue).

**Figure 2 pathogens-12-01304-f002:**
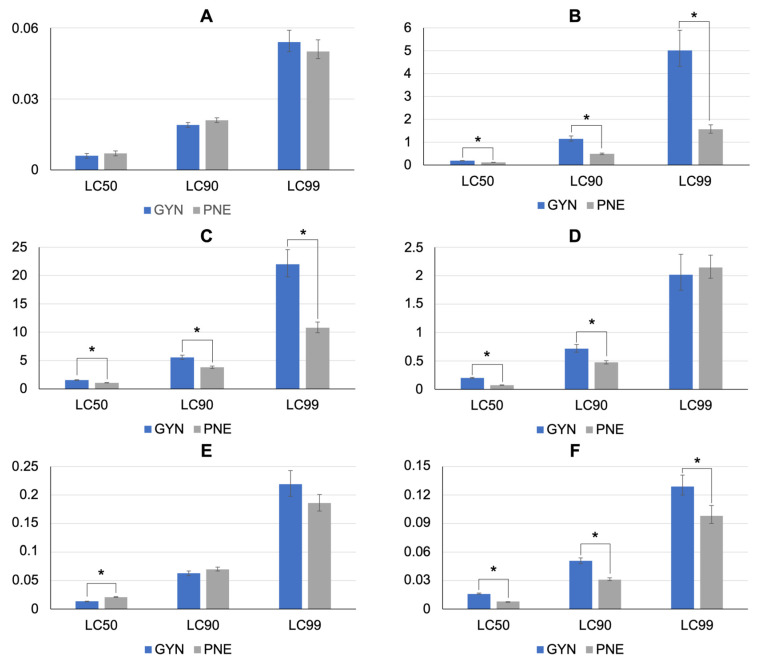
Lethal concentrations (LC_50_, LC_90_, and LC_99_) either in ppm (**A**–**D**) or percentage (**E**,**F**) for *Amblyomma sculptum* larvae from Goiás state, which were exposed to different acaricides using larval immersion tests. (**A**) Deltamethrin. (**B**) Cypermethrin. (**C**) Chlorfenvinphos. (**D**) Amitraz. (**E**) Chlorpyrifos + cypermethrin + citronellal. (**F**) Chlorpyrifos + cypermethrin + fenthion. Error bars represent 95% confidence intervals. An asterisk denotes statistically significant differences (*p* < 0.05).

**Figure 3 pathogens-12-01304-f003:**
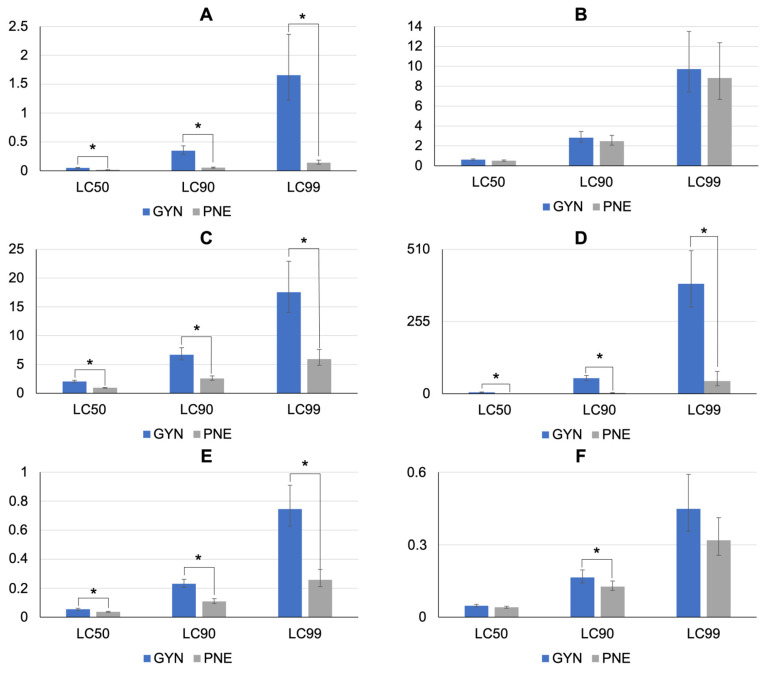
Lethal concentrations (LC_50_, LC_90_, and LC_99_) either in ppm (**A**–**D**) or percentage (**E**,**F**) for *Amblyomma sculptum* nymphs from Goiás state, which were exposed to different acaricides using nymphal immersion tests. (**A**) Deltamethrin. (**B**) Cypermethrin. (**C**) Chlorfenvinphos. (**D**) Amitraz (LC_50_ and LC_90_ of PNE are very low, thus not visible; for details, see Table 4). (**E**) Chlorpyrifos + cypermethrin + citronellal. (**F**) Chlorpyrifos + cypermethrin + fenthion. Error bars represent 95% confidence intervals. An asterisk denotes statistically significant differences (*p* < 0.05).

**Table 1 pathogens-12-01304-t001:** Commercial products used in this study, with trade name, manufacturer, chemical class, active ingredients, and concentrations (either in ppm or percentage), used in immersion tests with larvae and nymphs of *Amblyomma sculptum*.

Trade Name	Manufacturer	Chemical Class	Active Ingredient	Concentrations Used for Larvae/Number of Concentrations	Concentrations Used for Nymphs/Number of Concentrations
Butox^®^	MSD Animal Health	Synthetic pyrethroid	Deltamethrin	0.002 to 50 ppm/16	0.006 to 3.125 ppm/10
Barrage^®^	Zoetis Animal Health	Synthetic pyrethroid	Cypermethrin	0.018 to 150 ppm/14	0.073 to 9.370 ppm/8
Supokill^®^	UCBVet Animal Health	Organophosphate	Chlorfenvinphos	0.122 to 500 ppm/13	0.244 to 31.250 ppm/8
Triatox^®^	MSD Animal Health	Formamidine	Amitraz	0.008 to 250 ppm/16	0.008 to 250 ppm/16
Colosso^®^	Ourofino Animal Health	Synthetic pyrethroid + organophosphate + monoterpenoid	Cypermethrin + chlorpyrifos + citronellal	0.006 to 100%/15	0.012 to 6.25%/10
Colosso FC30^®^	Ourofino Animal Health	Synthetic pyrethroid + organophosphates	Cypermethrin + chlorpyrifos + fenthion	0.003 to 100%/16	0.012 to 6.25%/10

**Table 2 pathogens-12-01304-t002:** Lethal concentration (LC) and resistance ratio (RR) values for *Amblyomma sculptum* larvae from Goiás state, which were exposed to different acaricides using larval immersion tests.

Acaricide	Strain	LC_50_ (95% CI)	LC_90_ (95% CI)	LC_99_ (95% CI)	Slope	X^2^	RR_50_	RR_90_	RR_99_
Deltamethrin.	GYN	0.006 (0.005–0.007)	0.019 (0.018–0.020)	0.054 (0.050–0.059)	1.809	0.984	0.857	0.905	1.080
	PNE	0.007 (0.006–0.008)	0.021 (0.020–0.022)	0.050 (0.047–0.055)	3.986	0.994			
Cypermethrin	GYN	0.187 * (0.179–0.196)	1.147 * (1.042–1.270)	5.017 * (4.319–5.901)	1.630	0.989	1.612	2.360	3.214
	PNE	0.116 (0.112–0.120)	0.486 (0.455–0.522)	1.561 (1.397–1.761)	2.063	0.980			
Chlorfenvinphos	GYN	1.540 * (1.490–1.601)	5.544 * (5.196–5.938)	21.973 * (19.792–24.558)	1.747	0.991	1.453	1.457	2.038
	PNE	1.060 (1.024–1.097)	3.805 (3.607–4.026)	10.780 (9.900–11.807)	2.310	0.990			
Amitraz	GYN	0.200 * (0.191–0.210)	0.714 * (0.653–0.788)	2.017 (1.744–2.375)	2.316	0.963	2.703	1.509	0.941
	PNE	0.074 (0.071–0.077)	0.473 (0.445–0.503)	2.143 (1.955–2.359)	1.586	0.994			

CI: confidence interval; GYN: experimental farm strain; LC: lethal concentration (expressed in ppm); PNE: Emas National Park strain; RR: resistance ratio (calculated as the LC value of the GYN strain divided by that of the PNE strain). * Denotes statistically significant differences in LC values of GYN and PNE strains.

**Table 3 pathogens-12-01304-t003:** Lethal concentration (LC) and resistance ratio (RR) values for *Amblyomma sculptum* larvae from Goiás state, which were exposed to two acaricide combinations using larval immersion tests.

Acaricide	Strain	LC_50_ (95% CI)	LC_90_ (95% CI)	LC_99_ (95% CI)	Slope	X^2^	RR_50_	RR_90_	RR_99_
Chlorpyrifos + cypermethrin + citronellal	GYN	0.014 * (0.013–0.014)	0.063 (0.059–0.067)	0.219 (0.198–0.243)	1.934	0.978	0.667	0.900	1.177
	PNE	0.021 (0.021–0.022)	0.070 (0.067–0.074)	0.186 (0.172–0.201)	2.471	0.994			
Chlorpyrifos + cypermethrin + fenthion	GYN	0.016 * (0.016–0.017)	0.051 * (0.048–0.054)	0.129 * (0.120–0.141)	2.577	0.984	2.000	1.645	1.316
	PNE	0.008 (0.007–0.008)	0.031 (0.030–0.033)	0.098 (0.090–0.109)	2.11	0.983			

CI: confidence interval; GYN: experimental farm strain; LC: lethal concentration (expressed as %); PNE: Emas National Park strain; RR: resistance ratio (calculated as the LC value of the GYN strain divided by that of the PNE strain). * Denotes statistically significant differences in LC values of GYN and PNE strains.

**Table 4 pathogens-12-01304-t004:** Lethal concentration (LC) and resistance ratio (RR) values for *Amblyomma sculptum* nymphs from Goiás state, which were exposed to different acaricides using nymphal immersion tests.

Acaricide	Strain	LC_50_ (95% CI)	LC_90_ (95% CI)	LC_99_ (95% CI)	Slope	X^2^	RR_50_	RR_90_	RR_99_
Deltamethrin	GYN	0.051 * (0.045–0.058)	0.347 * (0.287–0.432)	1.655 * (1.225–2.364)	1.540	0.969	3.000	6.309	11.821
	PNE	0.017 (0.016–0.019)	0.055 (0.047–0.065)	0.140 (0.112–0.185)	2.568	0.971			
Cypermethrin	GYN	0.616 (0.547–0.691)	2.815 (2.378–3.436)	9.717 (7.419–13.494)	1.942	0.972	1.187	1.138	1.101
	PNE	0.519 (0.462–0.583)	2.474 (2.069–3.051)	8.827 (6.676–12.362)	1.891	0.967			
Chlorfenvinphos	GYN	2.039 * (1.842–2.253)	6.672 * (5.786–7.889)	17.534 * (14.069–22.911)	2.490	0.974	2.195	2.587	2.957
	PNE	0.929 (0.846–1.018)	2.579 (2.260–3.018)	5.929 (4.839- 7.611)	2.890	0.977			
Amitraz	GYN	4.943 * (4.293–5.658)	54.718 * (46.847–64.655)	388.497 * (306.422–505.384)	1.227	0.978	56.170	20.006	8.642
	PNE	0.088 (0.070–0.110)	2.735 (2.087–3.732)	44.954 (28.005–78.945)	0.859	0.952			

CI: confidence interval; GYN: experimental farm strain; LC: lethal concentration (expressed in ppm); PNE: Emas National Park strain; RR: resistance ratio (calculated as the LC value of the GYN strain divided by that of the PNE strain). * Denotes statistically significant differences in LC values of GYN and PNE strains.

**Table 5 pathogens-12-01304-t005:** Lethal concentration (LC) and resistance ratio (RR) values for *Amblyomma sculptum* larvae from Goiás state, which were exposed to two acaricide combinations using nymphal immersion tests.

Acaricide	Strain	LC_50_ (95% CI)	LC_90_ (95% CI)	LC_99_ (95% CI)	Slope	X^2^	RR_50_	RR_90_	RR_99_
Chlorpyrifos + Cypermethrin + Citronellal	GYN	0.056 * (0.049–0.063)	0.232 * (0.207–0.263)	0.746 * (0.627–0.911)	2.064	0.949	1.474	2.128	2.880
	PNE	0.038 (0.034–0.042)	0.109 (0.096–0.128)	0.259 (0.212–0.330)	2.796	0.988			
Chlorpyrifos + Cypermethrin + Fenthion	GYN	0.048 (0.043–0.054)	0.165 * (0.142–0.196)	0.448 (0.356–0.591)	2.407	0.987	1.171	1.299	1.409
	PNE	0.041 (0.037–0.046)	0.127 (0.111–0.150)	0.318 (0.257–0.412)	2.623	0.986			

CI: confidence interval; GYN: experimental farm strain; LC: lethal concentration (expressed as %); PNE: Emas National Park strain; RR: resistance ratio (calculated as the LC value of the GYN strain divided by that of the PNE strain). * Denotes statistically significant differences in LC values of GYN and PNE strains.

## Data Availability

Not applicable.

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
