# Peer review of "Susceptibility of *Amblyomma sculptum*, Vector of *Rickettsia rickettsii*, Ticks from a National Park and an Experimental Farm to Different Synthetic Acaricides"

_pathogens, 2023, doi:10.3390/pathogens12111304_

Round 1

Reviewer 1 Report

Comments and Suggestions for Authors

The manuscript is about susceptibility of two populations of Amblyomma sculptum to different synthetic acaricides. 

In introduction, kindly briefly add some background/previous studies where researchers used some of these acaricides to conduct tick susceptibility or/and resistance experiments.

In methods, authors mentioned A. sculptum tick strains. Did authors used molecular tools to differentiate between A. sculptum strains? 

2.4. Acaricides and lethal concentrations (2nd paragraph): Commercial products were directly purchased from consumer stores, and diluted in distilled water, following the manufacturer's instructions. Serial dilutions were performed to obtain the other concentrations. Kindly explain how did you prepare dozes/solutions of different acaricides or different concentrations of different acaricides for experiments? It is hard to follow by researchers.

In the nymphal immersion tests, a concentration range was optimized to achieve a mortality rate between 10% and 100%. Please explain how you optimized concentration range to achieve mortality rate.

The lethal concentration (LC) values for commercial products containing a single active ingredient were calculated based on the concentrations in parts per million (ppm). How? Formula?

2.6. Nymphal immersion tests (NITs), in this paragraph, kindly add reference/references.

Results: Please explain results more in detail. I would suggest if authors can add a graph/diagram to results which shows comparison between these acaricides in two tick populations. That will give quick overview of your study and results will be reader friendly and easy to follow.  

Discussion should be improved.

Conclusion is very short.

Authors contribution: Follow the journal style

Comments on the Quality of English Language

Kindly proofread carefully to correct English language errors.

Author Response

Dear Reviewer 1, 

Best regards, 

Reviewer 2 Report

Comments and Suggestions for Authors

1. I would rephrase this sentence in the introduction "Amblyomma sculptum is present in different South American countries, such as Argentina, Bolivia, Brazil, and Paraguay" by mentioning the species' preferred habitat in addition to the countries where its distribution has been confirmed.

2. The second paragraph is vague, and the first and second ideas are not well-connected.

3. The second and third paragraphs should be more connected, for instance,  the three types of acaricides are not included in the aim of the study. It is worth mentioning in the third paragraph that deltamethrin is a pyrethroid.

4. In 2.1. "(ICMBio Permit No. 70143-2)" should be before the word "authorized".

5. Last line of 2.2; "high level of environmental infestations by Amblyomma ticks has been previously detected in this area" clarify if this is by dragging/flagging or also in cattle.

6. In 2.3, second paragraph, how many adult ticks were collected approximately? Was only one individual rabbit used for all the ticks from the same strain?

7. In table 1. and sections 2.5 and 2.6, it is not established how many concentrations were used within the given ranges.

8. In Discussion, it will be interesting to discuss, together with the ideas already included, why the resistance to acaricides commonly used for cattle is more common than in the ones used for horses. Could it be related to the frequency of use? Are there other relevant factors for this?

9. The conclusions must be improved. 

Author Response

Dear Reviewer 2, 

Best regards, 

Reviewer 3 Report

Comments and Suggestions for Authors

The study evaluated the susceptibility of two Amblyomma sculptum ticks from Goiás state, Brazil, regarding acaricide exposure. Two strains were used in this study. One population (GYN strain) originated from an experimental farm regularly exposed to acaricides, while the other (PNE strain) was collected from a national park without acaricide exposure. Immersion tests were conducted on laboratory-reared larvae and nymphs using various acaricides in different concentrations The results showed differences in susceptibility between the acaricide-exposed (GYN) and unexposed (PNE) tick strains. PNE strain larvae and nymphs were susceptible to all tested acaricides. However, GYN strain larvae were tolerant to cypermethrin, and GYN strain nymphs were tolerant to deltamethrin, chlorfenvinphos, and a specific combination of pyrethroids and organophosphates. Notably, GYN strain nymphs exhibited resistance to amitraz, and this study marks the first report of A. sculptum nymphs with resistance to amitraz and tolerance to certain acaricides. The findings underscore the importance of understanding acaricide resistance in tick populations, particularly in public health, given A. sculptum's role as a vector for Rickettsia rickettsii in Brazil.

Strengths of paper

This paper is filling a knowledge gap since this is the first report of such a test that is been done in this area. It gives a comprehensive evaluation of the two strain populations and the strength of the study lies in its comprehensive evaluation of the susceptibility of two populations of Amblyomma sculptum ticks to various acaricides.

1.  By comparing two populations of A. sculptum from different environments (acaricide-exposed and unexposed), the study provides valuable insights into the potential development of acaricide resistance in ticks. This information can be used in future if scientists are targeting vectors for control of an infection.

2. The inclusion of different types of acaricides, including synthetic pyrethroids, organophosphates, formamidine, and combinations, enhances the robustness of the study. This diversity allows for a comprehensive assessment of resistance patterns.

3. The identification of resistance to amitraz and tolerance to specific acaricides in A. sculptum nymphs represents a novel contribution to the understanding of acaricide resistance in this species.

4. The tables and statistics were easy to follow and understandable.

5. The research question was addressed. 

6. Manuscript is clear and relevant to the field.

Weakness

1. They had two strains and a couple of treatments including their controls, the authors did not talk much about the percentages of larvae or nymphs for each treatment group, that is at the end of the studies how many of each treatment were tolerant, susceptible or resistant in each group.

2. The control should be added to your analysis, so we have a clear picture that the distilled water that was used in mixing your acaricides does not have any effect on the ticks as well. ( control/ distilled water effect)

3. Also what solution did you use for your dilution of acaricides, I believe is distilled water, hence the above comment describing if that has an effect on the acaricide.

3.  A paragraph about how your controls in water also behaved as compared to the acaricide-treated samples.

Comments on the Quality of English Language

The quality of English was good and easy to follow, I have a few corrections the authors have to make.

Abstract

1. Instead of which originated, this can be replaced with originating from adult ticks

2. Replace "in the above mentioned areas- with areas mentioned above

3. I believe the authors have done enough review to know this is the first report. Instead of the word "To our knowledge", be affirmative with your words i.e. This is the first report of such a work. it gives a sense of credibility to the article. the last paragraph To our knowledge, this is the first report of. The same statement is seen in the discussion kindly make changes to it.

A. sculptum nymphs with resistance to amitraz and tolerance to deltamethrin, chlorfenvinphos, and

the combination of cypermethrin, chlorpyrifos, and citronellal.

 this statement is also found in your discussion, kindly change them

Introduction

"Unlike what happens with these ticks, studies regarding the chemical control and acaricide resistance in A. sculptum are scanty "

The word scanty here can be changed to limited or any synonym preferably

Discussion

4. This statement below is hard to follow can you rewrite the sentence 

"However, in a study in five different municipalities

in the state (i.e., Caldas Novas, Hidrolândia, Goiás, Terezópolis and Goiânia), Freitas et al. [16] reported resistance to deltamethrin in adults and probably resistance to

deltamethrin, cypermethrin + piperonyl butoxide, amitraz and permethrin"

The statement probably may not be a good fit here. what was the conclusion of the article?

5. The authors compared what similar results other researchers got, instead of using the phrase in line with, the authors can simply say the results they got confirm what has been seen previously

Author Response

Dear Reviewer 3, 

Best regards, 

Round 2

Reviewer 1 Report

Comments and Suggestions for Authors

Authors complied with almost all comments. 

Only one comment, kindly check data and statistical analysis, in some cases error bars are large.

Comments on the Quality of English Language

Kindly proofread manuscript carefully.

Author Response

Dear Reviewer 1, 

Best regards, 
